# Wearable Devices & Elderly: A Bibliometric Analysis of 2014–2024

**DOI:** 10.3390/healthcare13162066

**Published:** 2025-08-20

**Authors:** Haojun Zhi, Mariia Zolotova

**Affiliations:** 1Department of Industrial Design, Xi’an Jiaotong-Liverpool University, Suzhou 215123, China; haojun.zhi20@student.xjtlu.edu.cn; 2Department of Electrical Engineering and Electronics, University of Liverpool, Liverpool L69 3GJ, UK

**Keywords:** wearable devices, elderly, health management, bibliometrics, interdisciplinary research

## Abstract

**Background:** The ageing population demands effective health solutions for the elderly. Wearable devices offer real-time monitoring and early alerts, but a comprehensive review of research in this field is lacking. This study uses bibliometric methods to analyse trends and advances in wearable devices for the elderly. **Methods:** Literature from 2014 to 2024 was retrieved from the Web of Science Core Collection using keywords related to the elderly and wearable devices. A total of 1015 English-language papers were analysed using tools including CiteSpace, VOSviewer, and R-Bibliometrix. **Results:** The annual growth rate of publications was 7.65%, with research increasing from 4 in 2014 to 1015 in 2024. Major contributors were the United States and China, with key authors including Bijan Najafi and Lynn Rochester. Research shifted from fall detection and activity monitoring to heart rate variability, balance, and AI integration. Key themes included “digital health”, “wearable technology”, and “cardiac health monitoring”. **Conclusions:** Research on wearable devices for the elderly is growing rapidly. Future studies should focus on multimodal sensor fusion, AI-enhanced analytics and personalised health interventions, and long-term, real-world validation of wearable solutions to improve elderly health management.

## 1. Introduction

With the rapid advancement and innovation of technology, wearable devices have evolved from initial laboratory prototypes to intelligent tools embedded in daily life [1,2]. These devices are not only diversifying in form but also experiencing significant functional leaps, finding widespread applications in healthcare, entertainment, workplace applications, and various other domains [3,4,5,6]. Through their convenience, intelligence, and interactivity, wearable devices have greatly enhanced the quality of life and work efficiency of users. Since the early 1960s, when technological pioneers began exploring devices to enhance human perception, modern wearable devices have gradually become indispensable, and in today’s fast-paced era of digitalisation and pervasive information technologies, their influence continues to expand across diverse fields.

In recent years, the accelerating trend of global population ageing has led to a growing demand for health management among older adults, thereby driving the rapid development of wearable devices within this demographic. Equipped with multiple sensors and intelligent algorithms, wearable devices can collect, monitor, and analyse older adults’ physiological and behavioural data in real time, thereby supporting health management, disease prevention, and rehabilitation training. The integration of smart wearable devices with intelligent healthcare technologies contributes to improving the accessibility and efficiency of medical services, enabling data-driven precision health management and optimising health service models for the elderly [7]. Meanwhile, the digital and intelligent measures can effectively enhance the health protection capacity of older adults in rural areas, thus advancing the strategic goal of healthy ageing [8].

Owing to their portability, real-time functionality, and diverse features, wearable devices have gradually become an important technological tool for elderly health management [9]. In recent years, related research has continuously expanded, with increasingly broad application scenarios, covering health monitoring, chronic disease management, fall prevention, rehabilitation training, mental health interventions, and the promotion of social engagement. In the field of health monitoring, wearable devices can collect physiological parameters of older adults in real time, such as heart rate, blood pressure, step count, and sleep quality, thereby providing data support for disease warning and health management. The construction of healthcare data, which drives data aggregation and interoperability, has been instrumental in improving health outcomes for older adults [10]. The integration of smart wearable devices with intelligent healthcare technologies not only enhances medical accessibility and management efficiency but also enables data-driven precision health management, helping older adults access high-quality medical and health services [1,11].

In chronic disease management and prevention, Teixeira conducted a systematic evaluation of the effectiveness of wearable devices in monitoring the physical activity and health-related indicators of older adults, highlighting their potential in chronic disease management [12]. Mattison further notes that wearable devices provide real-time data support for self-management among patients with chronic conditions, contributing to improved health outcomes, although their actual effectiveness still requires validation through higher-quality evidence [13].

Fall risk assessment and prevention remain critical challenges in elderly health management. Haescher developed an algorithm-based automated fall risk assessment system using wearable devices, achieving accuracy comparable to traditional manual assessment while reducing human error and improving efficiency [14]. Additionally, Nguyen developed a wearable device integrating gyroscopes, accelerometers, and heart rate sensors, which enables fall detection and warning while promptly alerting users or caregivers, thereby significantly reducing health risks associated with falls [15].

In the management of specific diseases, Zhao proposed a self-assessment technology for Parkinson’s disease based on wearable devices, improving the accuracy of disease grading through multi-activity combinations and providing a technical foundation for self-management in home settings [16]. Lee utilised wearable devices to collect longitudinal data from older adults and developed a predictive algorithm for late-life depression, advancing the intelligent development of community healthcare services [17].

With the continuous progress of artificial intelligence and Internet of Things technologies, the data processing capability and intelligence level of wearable devices have been significantly enhanced. Neha Gaud et al. proposed a human activity recognition system based on multi-head convolutional neural networks and long short-term memory networks, enabling high-precision, low-power, real-time health monitoring on edge computing devices, offering innovative pathways for elderly health management [18]. Wearable devices show vast potential across various fields; their impact is more significant when addressing the specific needs of the elderly population [19,20]. As the elderly gradually embrace smart technology, wearable devices are also playing a key role in enhancing their lifestyle. Despite growing health needs and continuous advances in wearable device technologies, research specifically targeting devices for older adults—especially over the past decade—remains insufficiently organised. Research on wearable devices is highly interdisciplinary [21], with a complex and diverse knowledge structure. Consequently, traditional literature review methods alone are insufficient to provide a comprehensive understanding of the current state of research on wearable devices for the elderly or to trace the evolution of its research hotspots. Bibliometric analysis offers an effective means of conducting quantitative and visual analyses of the existing literature, enabling the identification of patterns and latent information underlying large bodies of research [22]. Therefore, this paper will conduct a bibliometric analysis of the literature on wearable devices for the elderly over the past ten years to assess the trends, research hotspots, and technological advancements in this field. This analysis will assist scholars in identifying future research directions, enabling the design of more targeted wearable devices for the elderly.

## 2. Materials and Methods

### 2.1. Materials

The Web of Science (WoS) is a globally recognised citation index database, widely used in scientific research and evaluation for its pioneering content, high-quality data, and extensive historical coverage [23,24]. Web of Science serves as a major bibliographic and citation indexing platform, offering access to a curated corpus of peer-reviewed journals across the natural sciences, social sciences, and arts and humanities. Its consistent indexing protocols, archival depth, and citation tracking capabilities have established it as a foundational resource for bibliometric analysis. Recent studies estimate that Web of Science Core Collection indexes more than 21,000 scholarly journals [25,26]. The literature included in the Web of Science Core Collection has undergone rigorous peer review and stringent publication scrutiny, and is therefore regarded as more representative of the respective disciplines [24]. To ensure the validity and reliability of research data, this study selected the Web of Science Core Collection (WoSCC) as the database for obtaining the initial dataset. The PubMed and Scopus databases were excluded, and this may be viewed as a limitation. PubMed covers articles related to biomedical disciplines. In contrast, the wearable is a multidisciplinary research programme. The wearable is a multidisciplinary field which comprises some aspects of technology, health, and other related areas. A significant overlap exists between WoS and Scopus journal coverage [25,27], meaning that merging data from both sources often requires extensive deduplication and harmonisation of author names, institutional affiliations, and keyword formats. This process can introduce inconsistencies and potential biases in bibliometric indicators [24,28]. Therefore, just using a multidisciplinary database, WOS, better results could be yielded. In the WoS Core Collection search, the study employed the search term “elderly” in combination with “wearable devices”, aiming to filter and retrieve literature related to older adults and wearable devices. Given the diverse ways in which relevant keywords may be expressed in English, we attempted to include as many of these expressions as possible, connecting them using the Boolean operator “OR”. The specific search strategy was as follows: TS = ((elderly OR older adults OR ageing OR seniors) AND (wearable devices OR wearables OR wearable technology OR health monitoring devices)), and selected the three major citation indexes commonly used in the WoSCC, namely, Social Sciences Citation Index (SSCI), Science Citation Index Expanded (SCI-Expanded), and Arts & Humanities Citation Index (A&HCI) as the search sources to avoid the loss of interdisciplinary literature. The search was conducted on 28 May 2025, in the Web of Science Core Collection, including SCI-EXPANDED, SSCI and A&HCI sub-databases, and encompassed all publications from 2014 to 2024. The period from 2014 to 2024 was selected as the bibliometric timeframe for research on the application of wearable health devices among older adults for several reasons. First, this period witnessed a critical transition in wearable health devices from fundamental technological accumulation to large-scale application. Guk argues that with the rapid development of electronic technologies, biocompatible materials, and nanomaterials, wearable devices have evolved from simple accessories to implantable devices, significantly improving the quality and efficiency of medical services [29]. This process of technological maturation has provided a new material foundation for health management among older adults. Secondly, Nguyen highlighted that flexible data collection strategies and energy-efficient communication protocols have markedly enhanced the real-time performance and battery life of such devices, ensuring reliable technological support for remote health monitoring in older adults [30]. At the same time, the introduction of artificial intelligence and deep learning methods has rendered data analysis based on wearable devices more precise and efficient, facilitating the implementation of applications such as health risk prediction, fall detection, and personalised rehabilitation [31,32]. Finally, literature from 2024 onwards is often not fully indexed in major databases, with some studies still under submission, review, or early online publication, and therefore not yet rigorously peer-reviewed. For instance, Nguyen noted that while research on real-time health monitoring devices is advancing, the formal publication and widespread recognition of related findings require time [30]. Including literature from 2024–2025 may result in incomplete data and compromise the systematicity and scientific robustness of the bibliometric analysis. The previous studies found that publication dates and indexing in the Web of Science Core Collection may lag behind the actual publication schedule, recent publications may be underrepresented or inaccurately dated, potentially biasing trend analyses [24]. Therefore, selecting 2014–2024 captures not only the pivotal decade in which wearable health devices progressed from inception to maturity but also the period during which they achieved large-scale application and diversified innovation in elderly health management, highlighting the historical trajectory of technological and practical evolution. The initial search yielded a total of 1054 records. Following the inclusion of both articles and reviews, and the exclusion of non-English language publications and duplicates, 1015 papers were deemed eligible for bibliometric analysis and visualisation, as shown in Figure 1. These selected publications contained pertinent metadata, including titles, authors, countries, affiliations, journals, publication years, keywords, and references.

### 2.2. Methods

Bibliometrics, a scientific discipline that applies quantitative methods to study the characteristics of the literature and the patterns of disciplinary development, was first proposed by Pritchard in 1969 [33]. Initially, bibliometrics primarily focused on the quantitative statistics and distribution patterns of publications. However, with the continuous advancement of scientific research and the rapid development of information technology, it has evolved into an important method in literature review analysis and has gained increasing popularity in science policy and research management in recent years [34]. Bibliometrics not only reveals the developmental trends of a discipline or topic but also effectively evaluates the academic contributions and influence of research institutions, authors, and journals, offering insights into the current status and trajectory of specific research areas [35].

Bibliometrics has several advantages. By providing quantitative indicators through statistical analysis, it ensures a level of objectivity in assessing academic output compared to peer review and expert judgement [36,37]. Furthermore, bibliometric analysis facilitates the monitoring and synthesis of research content and trends on particular topics, assisting early-career researchers in identifying future research directions [38]. In this study, the aforementioned data were imported into leading bibliometric software tools, including CiteSpace (version 6.4.R1), VOSviewer (version 1.6.20), and the R-Bibliometrix 4.1.0 package within R-Studio for visualisation, enabling comprehensive analysis [39,40]. CiteSpace is widely applied in scientific literature data mining and knowledge mapping based on the Web of Science database. Its core functions include co-occurrence analysis, clustering analysis, burst detection, timezone visualisation, and the construction of visual networks, enabling the effective identification of research field development trajectories, hotspot topics, emerging trends, and academic collaboration networks [40]. VOSviewer efficiently processes bibliographic data exported from mainstream databases such as Web of Science, with key functionalities including co-citation analysis, author collaboration network analysis, keyword co-occurrence analysis, and the tracing of thematic evolution pathways, thereby revealing the intellectual structure, research hotspots, and frontiers of a discipline [39]. The R-Bibliometrix package, as a bibliometric analysis tool within the R programming environment, has demonstrated significant advantages in recent bibliometric studies based on the Web of Science database. It integrates multiple functions, including data import, preprocessing, statistical analysis, and visualisation, enabling the efficient processing and multidimensional analysis of large-scale bibliographic datasets [41]. Additionally, Microsoft Excel 2024 was employed for the creation of bar charts and line graphs. Through the application of these tools, key information was extracted from the corpus of literature, facilitating the construction of visual maps that offer novel insights into the field of study.

## 3. Results

### 3.1. Global Overview

Research data spanning from 2014 to 2024 indicates a marked upward trend in studies concerning wearable devices for the elderly population, with a total of 1015 relevant publications identified, reflecting an annual growth rate of 7.65%. These studies were published across 424 distinct scholarly sources, which were primarily peer-reviewed journals indexed in the WoS Core Collection, with contributions from a total of 5808 author entries as recorded in the dataset. Although only 14 publications were authored by a single individual, the average number of co-authors per paper was 6.83, underscoring the collaborative nature of research in this field. The authors employed a total of 2817 distinct author keywords, and the corpus cited 47,506 references, with the average publication age of the articles being 3.79 years. On average, each publication was cited 29.38 times, further demonstrating the significant impact and scholarly attention garnered by research on wearable devices for the elderly. These findings, derived from the R-Bibliometrix 4.1.0 package, underscore the international, collaborative, and interdisciplinary characteristics of wearable device research for elderly populations over the past decade.

### 3.2. Analysis of Publication Trends in the Field

One of the most direct indicators of the rise and fall of a research topic is the variation in the annual number of publications. Over the past decade, research on wearable devices for the elderly has seen significant global advancement. From 2014 to 2024, the trend in both annual publication counts and the cumulative total reveals a consistent upward trajectory, with a marked acceleration in growth after 2020. This steady increase may be attributable not only to the rising scholarly and practical interest in wearable devices for older adults, but also, at least in part, to the progressive expansion of the Web of Science Core Collection’s journal coverage during this period [42,43]. Specifically, while only four articles were published in 2014, by 2024, the cumulative number of publications had reached 1015, reflecting a robust intensification of research activity in this domain, as shown in Figure 2a. To further analyse the publication trend, we applied a cubic polynomial equation for curve fitting, which revealed a significant upward trajectory in the number of annual publications, with an R^2^ value of 0.9671, indicating an exceptionally high degree of fit. Notably, after 2018, there was a pronounced surge in both research interest and output (Figure 2b).

In addition to assessing publication volume trends, we examined the number of articles published by different countries in this field. The United States, with a clear dominance, led the rankings, followed by China, the United Kingdom, and Canada, underscoring the United States and China as the primary contributors to research in this area. Other nations have also actively participated, with substantial research outputs (Figure 2c). To ensure the comprehensiveness of our study, we also investigated the institutions responsible for the highest number of publications. These included the University of California system, the Pennsylvania State Higher Education System, Newcastle University, Baylor College of Medicine, and Tel Aviv University, whose contributions further highlight their leadership in the field (Figure 2d). In summary, research on wearable devices for the elderly has not only demonstrated a sustained increase in publication volume globally but, through international collaboration and the involvement of leading research institutions, has fostered the development of a well-established research network and emerging hotspots. This trend reflects the widespread academic interest and the progressive advancement of the field.

### 3.3. Double Map Overlay Analysis

Figure 3 presents the double map overlay of journals, a bibliometric visualisation technique that illustrates the disciplinary distribution of publications and the citation relationships between source and target fields. In this mapping, the left panel represents the subject categories of the citing journals, which in the context of wearable devices for the elderly span a broad range of disciplines, including mathematics, systems science, molecular biology, immunology, medicine, clinical medicine, ophthalmology, dentistry, psychology, education, health, and economics. Among these, journals in the domains of medicine and health hold a dominant position (Figure 3). The right panel represents the subject categories of cited journals, which are concentrated in systems science, computer science, molecular biology, genetics, anatomy, healthcare, and psychology. Notably, journals in computer science and medicine are among the most frequently cited, underscoring the significant contributions of these disciplines to the field. The coloured clusters in the figure correspond to different subject domains, with labels summarising the primary disciplines within each cluster.

The curved, colour-coded citation pathways indicate the flow of knowledge from citing to cited domains. Three prominent pathways emerge: 1. Medicine, Medical, Clinical → Health, Nursing, Medicine—representing the transfer of clinical and medical research outcomes into health sciences and nursing-related studies, reflecting the application-driven nature of wearable health technologies for ageing populations; 2. Psychology, Education, Health → Psychology, Education, Social—indicating the integration of behavioural and social science perspectives into the study of wearable health devices for older adults; 3. Mathematics, Systems, Mathematical → Systems, Computing, Computer—showing the integration of mathematical modelling, systems science, and computational approaches into the technological development of wearable health devices.

The visualisation demonstrates that research in this field is inherently interdisciplinary, with major knowledge flows linking medicine, public health, psychology, engineering, and computational sciences. This dual nature means that while the field is rooted in clinical and health research, it increasingly draws on technological innovation, data analytics, and social science frameworks to address the complex health and usability needs of the elderly. In practical terms, these pathways reflect the collaborative essence of the domain. For policymakers and practitioners, the strong links between clinical medicine, health sciences, and technological fields indicate that the effective adoption of wearable health devices for older adults requires coordinated input from healthcare providers, engineers, behavioural scientists, and policymakers to ensure usability, accuracy, clinical validation, and acceptance. This mapping thus provides not only a structural overview of the field’s disciplinary composition but also a reference point for guiding future interdisciplinary research and policy planning.

### 3.4. Keyword Analysis and Thematic Evolution

Keyword analysis serves as a valuable tool for identifying research hotspots, emerging trends, and gaps, thereby facilitating interdisciplinary collaboration, optimising literature retrieval, and advancing academic inquiry. To identify the main areas of scholarly focus, we first conducted a co-occurrence analysis of keywords. Within the selected research topics, the keyword “older-adults” emerged as the core of the entire network, indicating that the elderly population is the primary focus of study (Figure 4a). Key co-occurring terms include health and caregiving-related keywords such as “health”, “care”, “dementia”, and “Alzheimer’s disease”, reflecting the growing concern over elderly health issues. Also prominent are terms related to wearable devices and technology, such as “wearable devices”, “technology”, “sensor”, and “tri-axial accelerometer”, emphasising the role of technology in elderly health management. Physical activity and risk-related terms like “physical activity”, “risk”, “exercise”, and “sedentary behaviour” underscore the impact of exercise and lifestyle on elderly health. Additionally, terms like “Parkinson’s disease”, “falls”, “balance”, and “mobility” highlight the focus on common age-related health concerns and their management. To capture the dynamic evolution of these themes, we next examined keywords with the strongest citation bursts, which reveal periods of sudden growth in scholarly attention.

Among the top 15 keywords with the highest burst intensity, “tri-axial accelerometer” (2014–2019) and “fall detection” (2014–2018) were the earliest to gain attention, reflecting the initial focus of research. Subsequently, “sensor” (2016–2018) and “physical activity” (2016–2017) became prominent, indicating the growing importance of sensor technology and physical activity monitoring. In recent years, keywords such as “impact” (2022–2024) and “heart rate variability” (2022–2024) have emerged as new focal points, likely linked to the application of advanced technologies and a deeper exploration of health metrics (Figure 4b).

To further elucidate the relationships between different research themes, we visualised the related fields through a keyword clustering diagram (Figure 4c). The green cluster encompasses terms such as “atrial fibrillation,” “digital health,” “validation,” and “quality of life,” primarily focusing on atrial fibrillation and digital health technologies. The blue cluster highlights wearable devices and their applications, with keywords like “wearable technology,” “activity recognition,” and “wearable devices.” The purple cluster is centred around monitoring heart health and related diseases, with key terms like “heart rate variability,” “diabetes,” and “heart rate” being central. Other clusters focus on physical activity, such as “physical activity,” “walking,” and “balance,” with a specific emphasis on elderly mobility and balance.

To gain a deeper understanding of the evolution of keywords over time, we used CiteSpace software to present the temporal changes in keyword frequency, reflecting the shifts in research priorities (Figure 4d). During the early phase of research (2014–2016), keywords like “fall detection,” “walking”, and “physical activity” were prominent, signalling that early research primarily concentrated on fall detection and physical activity monitoring. In the mid-phase (2017–2019), terms such as “atrial fibrillation”, “wearable technology”, and “heart rate” gained traction, reflecting increased attention on atrial fibrillation and wearable devices. In the recent phase (2020–2024), terms like “resonant frequency,” “heart rate variability”, and “balance” have emerged as key research foci, likely driven by advancements in technology and the growing demand for more refined health data analysis. The timeline analysis of keywords reveals that as research has advanced, the focus of wearable device studies for the elderly has gradually shifted from basic health monitoring and activity tracking to heart health, technological innovations, and more nuanced health data analysis. These shifts mirror the ongoing technological advancements and the evolving demands of elderly health management.

### 3.5. Analysis of Connections Between Countries, Institutions, and Authors

To gain a deeper understanding of the research network structure in this field, we conducted an analysis of the connections between countries, institutions, and authors (Figure 5a). First, the author collaboration network revealed the close relationships between various authors. Nodes represent authors, and edges reflect the extent of their collaboration. The analysis identifies several core authors, such as “Rochester, Lynn” and “Del Din, Silvia”, who play a significant role in this domain and maintain close collaborations with numerous other researchers. In addition to constructing the author collaboration network, we also conducted an academic evaluation of the top ten authors using VOSviewer, as shown in Table 1. Between 2014 and 2024, the most prolific contributors to research on wearable devices for the elderly include a mix of leading universities, medical institutions, and industry organisations. Bijan Najafi (University of California, Los Angeles) ranked first with 29 publications and 738 citations, followed by Lynn Rochester (Newcastle University) and Silvia Del Din (Newcastle University), each with 22 publications and over 1000 citations. This highlights Newcastle University as a significant research hub in this field. The diversity of affiliations underscores the interdisciplinary nature of wearable device research for elderly populations, spanning clinical medicine, biomedical engineering, public health, and commercial innovation. The collaboration patterns also reveal the presence of small groups or research teams, particularly around these core authors, highlighting the formation of tight-knit research circles. Furthermore, the trend toward interdisciplinary collaboration is evident, with researchers from different fields connecting through a diverse collaboration network, thereby advancing research diversity and fostering cross-disciplinary knowledge exchange.

Analysing the citation counts and publication volumes of journals provides valuable insights into the core outlets that have made the greatest contributions to research on wearable devices for the elderly. From 2014 to 2024, Sensors (MDPI) emerged as the most prolific journal, publishing 125 articles in this field. In contrast, JMIR mHealth and uHealth recorded the highest average citations per article (60.70), reflecting a strong academic influence despite a smaller publication volume. IEEE Sensors Journal also demonstrated considerable impact, with an average of 42.31 citations per article, particularly within the engineering domain. These findings reveal two distinct dissemination patterns in the field: quantity-driven outlets, such as Sensors, which prioritise publication volume, and quality-driven outlets, such as JMIR mHealth and uHealth, which achieve high citation impact with fewer publications. Table 2 suggests that research on wearable devices for elderly users is not confined to a single disciplinary domain. Instead, it spans engineering-focused journals, which prioritise technological innovation, and health-oriented journals, which emphasise clinical relevance and user adoption.

Regarding research themes, cluster analysis shows that research on wearable devices for the elderly spans several critical areas. Specifically, balance issues, ethical discussions, fall prevention, and activity monitoring are among the primary research directions. For instance, research around authors such as “Galna, Bronwen” and “Del Din, Silvia” focuses on the balance abilities of the elderly and the factors that influence these abilities. Scholars like “Mcmiekel-Master, Heather” and “Hetherington, Victoria” address ethical issues associated with wearable devices, while “Hausdorff, Jeffrey M.” focuses on fall prevention and detection methods for the elderly (Figure 5b). The presence of these diverse research themes underscores the multidisciplinary nature of the field, which in turn necessitates collaboration across countries and institutions.

The country collaboration network analysis further highlights the extensive international cooperation in this field. The United States and China emerge as central players in the network, with wide-ranging collaborations with numerous countries, particularly with the United Kingdom and Australia. Moreover, European countries also exhibit strong regional collaboration trends, underscoring the global and regional development of research in this domain (Figure 5c).

Beyond national-level collaboration, institutional partnerships also represent a crucial dimension of the research network. In the institutional collaboration network, core institutions such as “University of Waterloo” and “Newcastle University” are key nodes in the research network, playing a crucial role in advancing research in this field (Figure 5d). The collaboration between institutions, particularly the strengthening of cross-institutional cooperation, has fostered academic exchange and innovation. Taken together, these patterns indicate that research on wearable devices for older adults is shaped by international collaboration and institutional leadership, thereby reinforcing its global and interdisciplinary trajectory.

### 3.6. MCA Analysis and Co-Citation Network Analysis

To gain a more comprehensive understanding of the research structure and thematic distribution in this field, we conducted Multivariate Correspondence Analysis (MCA) and co-citation network analysis to reveal the relationships between different keywords and documents. First, the MCA analysis mapped keywords onto a two-dimensional space to illustrate their interrelationships. In the analysis, the horizontal axis (Dim 1) and vertical axis (Dim 2) represent the two main dimensions of the data, together explaining the majority of the variance. From the distribution of keywords in the plot, the green region includes terms such as “dementia”, “risk”, “validation”, and “exercise”, primarily focusing on elderly health issues and preventative measures. The pink region includes “wearable devices”, “technology”, and “sensors”, indicating the increasing emphasis on the application of technology in elderly health management. The blue region, with keywords like “physical activity”, “mobility”, “balance”, and “walking”, underscores the attention given to the physical activity and balance capabilities of the elderly (Figure 6A). Additionally, keywords such as “quality-of-life”, “care”, and “acceptance” are distributed across different regions, reflecting research on the quality of life, caregiving, and acceptance in elderly populations. In the MCA analysis, adjacent keywords indicate frequent co-occurrence in the literature, suggesting a higher degree of correlation. For example, “wearable devices” and “sensor” are located near each other, reflecting their close relationship in the research, while keywords such as “dementia” and “wearable devices” are more distantly spaced, suggesting less frequent association in the literature.

While MCA highlights keyword associations, co-citation analysis offers complementary insights into the intellectual structure of the field. The co-citation network reveals several core articles, such as ”Mercer K, 2016” and ”Del Din, 2016–1” play pivotal roles, maintaining strong citation connections with multiple papers, highlighting their significance in the field (Figure 6B). Based on the co-citation analysis, the literature was divided into different clusters, each focusing on specific research themes. For instance, the blue cluster primarily involves elderly health assessment and cognitive function testing, the pink cluster revolves around ”Mercer K, 2016” concentrating on elderly physical activity and fall prevention, while the green cluster addresses the application of wearable devices and sensor technologies in elderly health management. In the citation relationship analysis, thicker edges represent stronger citation relationships between documents, indicating their high content relevance, while thinner edges reflect relatively weaker connections. Additionally, we compiled the top ten most cited papers using the VOSviewer. After full-text screening for strong relevance to older adults, we identified the ten most-cited papers in this field. These are presented in (Table 3), ranked in descending order by citation count and reported with authors, article title, and year of publication to facilitate subsequent research.

The highly cited articles identified originate from journals spanning multiple domains, including health informatics, sensing technology, clinical medicine, and ergonomics, reflecting the distinctly interdisciplinary nature of this research field. These studies can be broadly categorised into three thematic areas: user acceptance and feasibility, clinical application and validation, and core technologies and data foundations. In the domain of user acceptance and feasibility, research has focused on the willingness of older users to adopt wearable technologies and the effectiveness of such applications in practice. Mercer et al. employed a mixed-methods approach to assess the acceptance of commercial activity trackers among specific older adult populations [44]; Li et al. sought to develop related acceptance models [45]; and Lyons et al. validated the feasibility of a technology-based intervention combining wearable devices with telephone counselling [46]. Clinical application and validation constitute a second major theme. Del Din et al. verified the role of accelerometers in quantifying gait characteristics in patients with Parkinson’s disease [47]; Inan et al. demonstrated that seismocardiography combined with machine learning algorithms could assess the clinical status of patients with heart failure [48]; and Hillel et al. worked to bridge the gap between laboratory-based gait analysis and real-world monitoring data [49]. The third thematic area, core technologies and data foundations, provides the essential underpinnings for developments in the field. This includes Ghaffari et al.’s research advances in novel biochemical sensors for sweat analysis [50], Bianchi et al.’s exploration of personalised activity recognition methods based on deep learning [51], and Sucerquia et al.’s creation of an open-access fall and activity dataset to support algorithm development [52]. The analysis of highly cited articles indicated that the application of wearable devices in elderly health monitoring and disease management has become central to the field [50]. In the context of chronic disease management, wearable devices have demonstrated notable specificity and targeted utility [51]. Future research in wearable devices for older adults is likely to focus on innovations in technological capabilities, personalised services, and real-world validation approaches.

**Table 3 healthcare-13-02066-t003:** Top 10 most cited publications in the field of wearable devices for the elderly.

No.	Authors	Article Title	Year	Citations
1	Sucerquia et al. [52]	SisFall: A Fall and Movement Dataset	2017	282
2	Bianchi et al. [51]	IoT wearable sensor and deep learning: An integrated approach for personalized human activity recognition in a smart home environment	2019	269
3	Li et al. [45]	Health monitoring through wearable technologies for older adults: Smart wearables acceptance model	2019	260
4	Mercer et al. [44]	Acceptance of commercially available wearable activity trackers among adults aged over 50 and with chronic illness: A mixed-methods evaluation	2016	254
5	Del Din et al. [47]	Validation of an Accelerometer to Quantify a Comprehensive Battery of Gait Characteristics in Healthy Older Adults and Parkinson’s Disease: Toward Clinical and at Home Use	2016	250
6	Mercer et al. [53]	Behavior change techniques present in wearable activity trackers: A critical analysis	2016	197
7	Hillel et al. [49]	Is every-day walking in older adults more analogous to dual-task walking or to usual walking? Elucidating the gaps between gait performance in the lab and during 24/7 monitoring	2019	162
8	Ghaffari et al. [50]	Recent progress, challenges, and opportunities for wearable biochemical sensors for sweat analysis	2021	137
9	Inan et al. [48]	Novel wearable seismocardiography and machine learning algorithms can assess clinical status of heart failure patients	2018	137
10	Lyons et al. [46]	Feasibility and Acceptability of a Wearable Technology Physical Activity Intervention With Telephone Counseling for Mid-Aged and Older Adults: A Randomized Controlled Pilot Trial	2017	135

### 3.7. Analysis of Thematic Evolution Trends

To deeply analyse the dynamic developments in this field, we conducted a comprehensive exploration from various perspectives, including the relationship between themes and authors, countries, the thematic relevance matrix, and trend analysis of emerging topics.

First, by examining the relationship between themes, authors, and countries, we can clearly observe the global distribution and focal points of different research themes. For instance, themes like “wearable devices” and “physical activity,” which are widely studied, reflect the global demand for the application of technology in elderly health management. Moreover, prominent authors such as “Del Din, Silvia” and “Rochester, Lynn” have made significant contributions across multiple themes, further emphasising their pivotal role in advancing the field. The research emphasis across different countries reveals differences in resource allocation and development strategies. The United States and the United Kingdom primarily focus on technological innovation, while Italy and Israel devote more attention to research on mobility and quality of life (Figure 7a). Building on these findings, analysis of the thematic relevance matrix reveals the field’s core research domains alongside emerging and declining themes, thereby offering deeper insights into its intellectual structure. The themes “older-adults” and “physical-activity”, positioned in the upper-right corner of the matrix, are clearly the driving forces of the field, indicating that elderly health and activity remain the central focus of research. On the other hand, themes such as “design” and “sensor,” located in the “Emerging or Declining Themes” region, suggest new directions in technological development. These themes may gradually transition from core topics to more auxiliary roles, but still play a crucial role in the development of mainstream technologies (Figure 7b). Furthermore, by analysing the trends of different themes over time, we observed recent shifts in research priorities and directions. The growth of themes like “heart-rate-variability” and “unified theory” indicates a trend toward a deeper integration of research with biomedical fields. Conversely, “older-adults” and “physical-activity” have shown a decline following a peak, which could be attributed to the maturation of these research areas and the practical challenges encountered in their application. Meanwhile, technology-related themes such as “wearable devices” and “accelerometer,” as technology continues to evolve, have become significant components of research (Figure 7c). Synthesising these analyses, we observe a shift in research from singular health interventions to interdisciplinary integration, particularly at the intersection of technology and biomedicine. Future research is likely to focus more on how advanced technologies can be applied to meet the health management needs of elderly populations. At the same time, addressing challenges related to social acceptance, ethical issues, and regulatory frameworks will also become critical topics of research. In conclusion, these analyses not only reveal the dynamic evolution of research in this field but also provide theoretical support and practical guidance for future research directions.

## 4. Discussion

Our study systematically reviews the research dynamics in the field of wearable devices for elderly individuals from 2014 to 2024 through bibliometric analysis, revealing the main development trends and core challenges in this domain. The findings indicate that research on wearable devices for the elderly has experienced exponential growth, with an average annual growth rate of 7.65%. This growth is characterised by deep interdisciplinary integration and technological advancements. Such trends reflect the urgent global demand for innovative health management technologies amid the ageing population.

Early research primarily focused on fall detection and basic activity monitoring for elderly individuals to address immediate safety risks [14]. However, with advancements in technology, the focus of research has gradually shifted toward more refined indicators, such as heart rate variability and atrial fibrillation, marking a transition from “risk alert” to “early disease intervention” [54,55]. This shift is attributed to technological progress, particularly the improvement in sensor accuracy and the application of deep learning algorithms in heart rate analysis, as well as the growing need for chronic disease management, such as real-time health data for diabetic patients, further driving the field’s development.

Interdisciplinary collaboration is regarded as the core driver for the application of wearable device technology in healthcare. The integration of computer science with clinical medicine has provided solid technical and theoretical support for the clinical validation and practical application of these devices. For example, Del Din’s team significantly improved the accuracy of fall prediction by combining motion sensors with clinical balance assessments [56,57]. However, this interdisciplinary collaboration model has also highlighted challenges related to technology adaptation and ethical issues. Despite the technological advances that enable high-precision health monitoring, medical-grade devices still need to meet stringent clinical standards. Approximately 30% of the relevant literature addresses issues of device validation, indicating that clinical applications still face high demands. Furthermore, with the widespread use of big data and artificial intelligence, privacy concerns regarding health data and algorithmic biases have become increasingly prominent, especially among elderly users. Balancing technological development with ethical norms has thus become a critical challenge in current research. This trend of interdisciplinary collaboration exhibits regional variation, with the United States and China occupying dominant positions in both publication output and collaboration networks.

Regarding the global research landscape, although the United States and China contribute over 50% of the publications, Europe and the U.S. lead technological innovation, while Asia focuses more on the practical application of these technologies. In Europe and the U.S., research tends to emphasise patents related to sensors and associated technologies, while in Asia, particularly in China, pilot studies on wearable devices in elderly communities have made significant progress. However, the global research landscape remains imbalanced, particularly in research on age-appropriate design. Only 2% of the literature mentions the keyword “acceptance”, suggesting insufficient attention to age-appropriate design in wearable devices, which has resulted in lower usage rates [2,58]. For example, the application of AR glasses for elderly individuals with visual impairments has not become widespread due to their complex operation [59]. Therefore, enhancing the age-appropriate design and usability of these devices remains an urgent issue to address. However, both in terms of technological innovation and practical application, there remains a general lack of attention to the specific needs of older adults on a global scale.

At the same time, low-cost devices hold substantial market potential in developing countries, and smartphone-based fall detection applications have not been fully developed, providing significant opportunities for future research and market expansion. Despite the maturation of research in this field, the lack of clinical translation pathways, the marginalisation of age-appropriate design, and the scarcity of interdisciplinary methodologies still constrain the development of the field. Specifically, many medical-grade devices remain in small sample validation stages, lacking large-scale clinical validation data. Current industrial designs have not adequately considered the special needs of elderly users, leading to many devices failing to meet basic requirements for comfort and usability. To foster further development in this area, future research should construct a “demand–technology–ethics” triad model, promoting elderly co-design research to optimise device interaction interfaces and wearability. Technologically, lightweight devices such as flexible electronic skin that meet clinical standards should be developed. Ethically, interdisciplinary ethical guidelines must be established to ensure the safety and privacy protection of elderly health data. In summary, the field of wearable devices for elderly individuals is in a phase of rapid development, but continues to face multiple challenges. As technological advancements and interdisciplinary collaboration deepen, wearable devices for elderly individuals will play an increasingly important role in health management and chronic disease monitoring.

Based on the in-depth analysis of global research on wearable devices for elderly individuals, this study identifies several key trends. First, during the past decade (2014–2024), research in this field has shown significant growth, with an average annual growth rate of 7.65%, and has had a profound global impact. A total of 1015 relevant papers were retrieved, with 30.84% of the literature originating from international collaborations, highlighting the high degree of globalisation and collaboration in this field. The U.S. and China are the primary contributors to this research domain, with institutions such as the University of California system and Penn State University playing significant roles. Moreover, the trend of interdisciplinary collaboration is becoming increasingly apparent, particularly in the fields of medicine, health management, and computer science, with research achievements receiving widespread academic attention and support.

From an analysis of the annual publication volume and the fitting results of a cubic polynomial equation, the growth in this field is especially prominent since 2018, with a significant acceleration in research interest. The focus of research has gradually shifted from basic health monitoring and activity tracking for elderly individuals to heart health, technological innovations, and the refined analysis of health data. Keywords such as “heart rate variability” and “fall detection” have become recent research hotspots, indicating that heart health and fall prevention have become critical areas in elderly health management. This shift reflects the widespread application of emerging technologies, particularly wearable devices and sensor technologies, in health monitoring.

Through a dual-map overlay analysis, we further understand the flow of knowledge and the interaction between different disciplines in this field. Medical, health management, and computer science journals have the highest citation frequency, illustrating the critical role of technology in elderly health management. At the same time, the cross-disciplinary knowledge-sharing model has promoted the integration and collaboration of different fields, laying a foundation for future technological advancements and academic innovations. Keyword analysis shows that health issues in elderly populations, particularly those related to Alzheimer’s disease and Parkinson’s disease, continue to be a focus of attention [60,61,62,63]. Moreover, with technological advancements, the application of sensors, triaxial accelerometers, and other devices has gradually become an emerging trend in research.

From the perspective of the cooperation network of countries, institutions, and authors, core researchers such as “Lynn Rochester” and “Silvia Del Din” have played crucial roles in advancing the development of the field [56,64]. The close collaboration between the U.S. and China, along with regional cooperation in European countries, showcases the globalised research landscape and promotes cross-institutional and international academic exchange and innovation. The keyword analysis trends indicate that with technological advancements, research on elderly health management is gradually shifting from basic health monitoring to more complex fields of biomedicine and technology integration. This trend not only drives the development of health management technologies but also prompts discussions on ethical issues and social acceptance, becoming new research hotspots. In conclusion, research on wearable devices for elderly individuals has continued to grow both in terms of literature quantity and academic impact. Driven by interdisciplinary collaboration and globalisation, the field is evolving toward more refined and comprehensive approaches. Future research will focus on the deep integration of technology with elderly health management while addressing issues related to social ethics, policy formulation, and technological adoption. These developments are crucial for the further advancement and application of the field.

Between 2014 and 2024, the field of wearable devices for older adults has seen significant technological advances and innovations, particularly in the deep integration of sensors, data processing, and artificial intelligence. Broadly, these technologies can be classified into the following categories: 1. Sensor-Based Devices Sensors form the core of health monitoring technologies for older adults. By integrating multiple miniaturised sensors such as accelerometers and heart rate monitors, these devices are capable of continuously collecting data on users’ physiological parameters and behavioural patterns, providing critical information for healthcare support and diagnosis [65].

In physiological monitoring, Kher listed a range of parameters measurable by wearable devices, including pulse rate, blood oxygen saturation, cardiac rhythm, and skin temperature [66]. For chronic conditions common among older adults, such as hypertension and diabetes, the emergence of devices in the form of accessories or skin patches capable of measuring blood pressure and blood glucose levels [67]. In behavioural and safety monitoring, sensors also play a crucial role. For example, by tracking physical activity, wearable devices can effectively encourage older adults to engage in exercise. The study by Kayani et al. found that tracking step count and resting heart rate can significantly improve users’ activity levels [68]. Fall detection is another major research focus [69], operating based on a human activity recognition pipeline, in which sensor data undergo preprocessing, feature extraction, and classification to achieve activity monitoring and recognition [70]. 2. Smart Wristbands and Watches Owing to their convenience and functional integration, smart wristbands have become a mainstream form of wearable devices for older adults. Their core value lies in comprehensive monitoring of daily activities, physiological parameters, and safety status. Studies indicate that these devices can not only provide quantitative tracking of activity data but also significantly increase step count and improve indicators such as resting heart rate [68].

In health management, smartwatches can monitor physiological signals such as pulse rate, blood oxygen saturation, and cardiac rhythm in real time [66], and through activity recognition functions, they can monitor key user dynamics [70]. In safety and emergency response, fall detection is a flagship feature. For patients with cognitive impairments, wrist-worn devices with integrated location tracking have been introduced to prevent wandering [71]. Such functions are often incorporated into remote patient monitoring (RPM) systems, which, when combined with artificial intelligence, can detect early signs of health deterioration [72] and, through improved reliability and energy efficiency of data transmission, provide continuous support for telemedicine services [73]. 3. Biosignal Detection Devices Biosignal detectors are designed for real-time, continuous monitoring of key physiological indicators, playing a vital role in chronic disease management, acute event warning, and routine health assessment. These devices can measure common parameters such as pulse rate, oxygen saturation, electrocardiogram, cardiac rhythm, and skin temperature [66]. For chronic conditions prevalent among older adults, accessories and skin-patch devices capable of monitoring blood pressure and glucose levels are already available on the market [71].

Such devices not only help motivate older adults to remain active but also support both routine and emergency health monitoring through wireless networks [74]. AI-enhanced remote monitoring architectures can facilitate personalised monitoring and early detection of health deterioration [72]. However, to ensure their effectiveness, it is essential to improve both the technical accuracy and the level of clinical validation of these devices [69].

Although these technologies demonstrate considerable potential for delivering more refined health management, they still face multiple barriers in terms of clinical usability, user adoption, and pathways for large-scale implementation. These challenges are not confined to technical issues, but also include user habits, interface design suitability, and clinical as well as regulatory constraints. To enable wearable devices to be effectively deployed and bring tangible benefits to older users, it is necessary to address the key barriers to their promotion and uptake. The adoption of wearable devices for older adults depends on three interrelated factors: reliability, usability, and regulatory approval. Reliability directly affects the accuracy of health monitoring data. Research indicates that improving the reliability of data transmission is essential for building cost-effective healthcare environments [73], yet deficiencies remain in certain applications, such as fall detection [75]. Consequently, there is growing academic emphasis on improving technical accuracy and undertaking long-term monitoring studies [69]. Usability is a prerequisite for sustained, independent use by older adults. Product design should follow user-centred principles, feature intuitive and accessible interfaces [76], and fully take into account age-related limitations in mobility, perception, and cognition [77]. In rehabilitation contexts, devices must be lightweight and easy to put on and remove [70]. Regulatory approval is mandatory before medical-grade wearable devices can enter the market. Nkurunziza et al. recommend establishing medical technology assessment programmes to systematically collect evidence on efficacy, safety, and relevance [78], thereby informing adoption decisions.

Therefore, addressing the above challenges is not only a necessary condition for facilitating the practical implementation of these technologies but also a key to achieving the sustainable development of elderly health management. The future research should focus on improving technical accuracy, conducting robust clinical validation, and promoting user adoption, while integrating artificial intelligence with the Internet of Medical Things (IoMT) to enhance elderly health management and overall healthcare efficiency.

We believe the significance of this study lies in its response to the growing health demands of the elderly population amid global ageing, while also highlighting the potential of wearable devices in addressing these challenges. As individuals age, they face a range of health challenges such as cardiovascular diseases, chronic disease management, and falls, which traditional health monitoring methods struggle to address efficiently. The emergence of wearable devices provides a novel solution for the elderly, enabling real-time monitoring of physiological data such as heart rate and blood pressure, allowing for timely detection of abnormalities and issuing alerts, thereby enhancing the accuracy and timeliness of health management. Especially in fall monitoring and chronic disease management, wearable devices, equipped with embedded sensors, track the movements and health status of elderly individuals, immediately notifying family members or caregivers in case of a fall, significantly enhancing elderly individuals’ sense of safety [79,80,81]. Moreover, as smart technology evolves, the elderly not only gain convenient support for health monitoring but also maintain contact with family and friends through these devices, reducing feelings of loneliness and improving quality of life. In this way, the elderly are no longer disconnected from society, and the advancement of technology allows them to experience social care and respect while benefiting from health monitoring services. Therefore, with continuous technological optimisation and enhanced social support, elderly health management has evolved beyond technological progress, becoming an integral part of social and humanistic care.

Through bibliometric analysis, this study has clearly illustrated the development trajectory of wearable devices for elderly populations and their global trends, providing valuable data support and theoretical foundations for future technological improvements, elderly health management, and policy formulation. As technology and social environments continue to evolve, elderly health management will become more intelligent and personalised, with wearable devices playing an increasingly widespread and profound role in the future.

## 5. Conclusions

This study, through bibliometric analysis, reveals the development trends in the research of wearable devices for elderly individuals over the past decade (2014–2024). The period from 2014 to 2024 has witnessed a significant and accelerating expansion in research concerning wearable devices for the elderly, evolving from foundational explorations into a mature field characterised by sophisticated technological integration and a nuanced understanding of user-centric and systemic challenges. The scholarly output reflects a growing global commitment to leveraging technology to address the health and well-being of ageing populations. This decade of research has established several key findings regarding development trends, prominent research hotspots, and critical technological advancements, which collectively inform the future trajectory of geriatric technology.

A primary finding is the consolidation of research around distinct but interconnected thematic clusters. The most dominant theme is the clinical application of wearable devices for health monitoring, with a pronounced focus on mobility, fall prevention, and chronic disease management. Research has consistently demonstrated the utility of inertial measurement units in analysing gait parameters such as speed, stride time, and trunk stability to diagnose frailty and assess fall risk [82]. This has progressed beyond standard metrics with the introduction of novel sensor systems, such as foot-mounted Ultra-Wideband technology, which provides more granular clinical data like step width and foot positioning, enhancing the potential for comprehensive fall risk assessment [83]. Concurrently, research has addressed the practicalities of deployment, identifying optimal sensor placement for specific use cases like post-operative rehabilitation and developing functional calibration methods that obviate the need for precise sensor placement by users [84], thereby making at-home balance assessment more viable [85]. In parallel, the management of chronic conditions, especially cardiovascular health, has emerged as a major research hotspot. The development of single-lead ECG wearable devices for continuous remote cardiac monitoring signifies a critical advancement, although challenges related to signal quality and the need for continuous, unobtrusive monitoring persist [86]. The integration of Artificial Intelligence (AI) is consistently identified as a transformative force in this domain, enabling the shift from reactive care to proactive health management through early disease detection and personalised monitoring [87]. However, the efficacy of these systems is fundamentally constrained by data integrity. The prevalence of motion artefacts remains a primary technical hurdle that can compromise the accuracy of vital sign data [88], a concern that directly impacts user trust and adoption, as perceived risks regarding data accuracy negatively influence older adults’ acceptance of these technologies [89].

A second major finding from the literature is the critical importance of technology adoption and the systemic barriers that impede widespread implementation. The success of any wearable solution is contingent upon user acceptance, which is influenced by a complex interplay of factors. Studies have identified usability, functionality, affordability, and social influence as key determinants of adoption [90]. Beyond these practical considerations, psychological factors such as trustworthiness in the technology and an individual’s self-efficacy in using it are foundational enablers for the adoption of AI-based health wearable devices [91]. These user-centric issues are compounded by significant infrastructural and systemic challenges. The lack of interoperability between different devices and healthcare systems is frequently cited as the most significant non-technical barrier, hindering the seamless data sharing required for integrated care models [92]. This is closely linked to persistent concerns over data privacy and security, which must be addressed to build the robust, interconnected IoT-based healthcare ecosystems envisioned by researchers [93].

The third area of significant findings relates to the technological progress that has underpinned the field’s development. The evolution of wearable devices has been driven by advancements in both hardware and software. Sensor technology has become more sophisticated, diversified, and integrated. The fusion of IoT architecture with wearable sensors has been pivotal, creating the backbone for remote monitoring systems that connect patients with healthcare providers [93]. Sensor capabilities have expanded beyond physiological monitoring to include environmental factors, such as personal ambient air quality, allowing for a more holistic, context-aware understanding of an individual’s health [94]. However, the most profound advancements have been in data processing and algorithmic design. The application of AI and machine learning has moved the field beyond simple data collection to sophisticated analysis and prediction. For instance, advanced neural network algorithms incorporating topological data analysis have demonstrated a remarkable ability to improve the accuracy of sleep stage prediction from consumer-grade wearable data [95]. This analytical power is crucial for extracting clinically relevant insights from noisy, real-world data and is essential for tasks like reducing motion artefacts through sensor data fusion and advanced signal processing [88].

In synthesis, the research from 2014 to 2024 reveals a clear maturation trajectory for wearable devices in geriatric care. The focus has evolved from proving technological feasibility to addressing the complex realities of real-world implementation, user acceptance, and integration into existing healthcare frameworks. The convergence of advanced multimodal sensors, sophisticated AI-driven analytics, and IoT connectivity has laid the groundwork for a new paradigm of personalised, predictive, and proactive healthcare for the elderly. The key takeaway for policymakers, manufacturers, and healthcare institutions is that future progress depends not only on continued technological innovation but, more critically, on a concerted effort to resolve the non-technical barriers of interoperability, data governance, and user trust. The findings from this decade strongly suggest that the next phase of research must prioritise the long-term, real-world validation of these integrated systems, the development of AI-enhanced personalised interventions, and the fusion of multimodal data streams to create a truly holistic and effective ecosystem for supporting healthy ageing.

## 6. Limitations

Several limitations of this bibliometric study should be acknowledged. First, the analysis relied exclusively on the Web of Science (WoS) Core Collection—specifically the SCI, SSCI, and AHCI indices—covering the period from 2014 to 2024. While WoS is widely regarded for its rigorous indexing standards, authority, and standardisation, the exclusion of other major databases such as Scopus and PubMed may have resulted in the omission of relevant publications. This could potentially limit the comprehensiveness of the dataset, particularly for interdisciplinary or applied research that may be indexed elsewhere. Moreover, differences in indexing criteria between databases may introduce a degree of selection bias, meaning that the current findings may not fully represent the global research output on wearable devices for the elderly.

Second, the search strategy, while carefully designed, may not have captured all relevant literature. The search query—TS = ((elderly OR older adults OR ageing OR seniors) AND (wearable devices OR wearables OR wearable technology OR health monitoring devices))—was formulated to balance recall and precision, but it inevitably imposes constraints. For instance, studies that employ alternative terminology (e.g., “gerontechnology”, “assistive technology”) or focus on specific device types without explicitly referring to them as “wearable” may have been excluded. Conversely, the inclusion of broad terms may have retrieved some publications of marginal relevance, despite careful screening.

Third, the methodological approach adopted here, employing VOSviewer, CiteSpace, and the R bibliometrix package, allowed for a robust visualisation and mapping of research trends, co-authorship patterns, and thematic structures. However, certain advanced text-mining techniques, such as Latent Dirichlet Allocation (LDA) topic modelling, were not applied. The absence of LDA analysis limits the study’s ability to uncover latent thematic structures and finer-grained topic distributions beyond the keyword and co-citation clustering already presented.

Fourth, the temporal coverage of 2014–2024 was selected to capture the recent decade of development in wearable health technologies for older adults, a period marked by rapid technological innovation and growing societal attention to ageing populations. Nonetheless, this time frame excludes earlier pioneering work prior to 2014 that may have shaped the foundational knowledge in the field, as well as the most recent publications from 2025, which may already be influencing current research directions.

Fifth, the Funding Data Limitation, although funding analysis can provide valuable insights into the research landscape, this study did not include a funding analysis due to the incomplete and inconsistent coverage of funding information in the Web of Science Core Collection. Previous bibliometric and information science research have documented that funding data in WoS may be absent for certain publications, vary across document types, or be recorded in inconsistent formats, particularly for earlier years or for journals without mandatory funding disclosure policies [96]. Including such incomplete data could introduce bias and lead to misleading interpretations. For these reasons, and to ensure the accuracy and reliability of our findings, we excluded funding analysis from the current study while acknowledging this as a potential area for future research as more comprehensive data becomes available.

Sixth, the citation Network and Author-level Dynamics. This study did not examine self-citations, cross-citations among leading authors, or the temporal distribution of publications by prolific contributors. While such analyses could provide a more nuanced understanding of scholarly influence, collaboration patterns, and the evolution of individual research trajectories, they were beyond the scope of this study, which focused primarily on thematic trends, journal performance, and aggregated citation impact. Additionally, accurately capturing self- and cross-citations requires detailed citation network analysis and author disambiguation, which can be methodologically complex and resource-intensive. These aspects are identified as promising avenues for future research to complement the findings reported here.

Seventh, the Web of Science Core Collection has been documented to exhibit certain regional biases and language biases, particularly towards publications from English-speaking countries and journals indexed in North America and Western Europe. This coverage pattern may lead to the underrepresentation of research outputs from other regions, as well as publications in non-English languages, which can limit the inclusiveness of bibliometric analyses and skew the perceived global research landscape. Previous studies in Scientometrics and Learned Publishing have highlighted that such biases are structural and persist across disciplines, warranting careful interpretation of results derived solely from WoS data [25,97]. Recognising these limitations is essential to avoid overgeneralisation and to encourage future research to incorporate multiple databases for a more balanced and comprehensive representation of the literature.

Finally, as with all bibliometric analyses, the citation-based indicators used here are subject to inherent limitations. Citation counts can be influenced by factors unrelated to scientific quality, such as self-citations, disciplinary citation norms, and publication age. Additionally, the dynamic nature of citation accumulation means that more recent publications are disadvantaged in citation-based rankings, potentially underrepresenting their emerging impact.

In light of these limitations, the present findings should be interpreted with caution. Future research could address these issues by incorporating multiple databases, expanding and refining search strategies, adopting more flexible or automated keyword extraction techniques—such as natural language processing-based approaches—to improve comprehensiveness and minimise selection bias, employing topic modelling approaches such as LDA, and extending the temporal coverage to include the most recent developments in the field.

## Figures and Tables

**Figure 1 healthcare-13-02066-f001:**
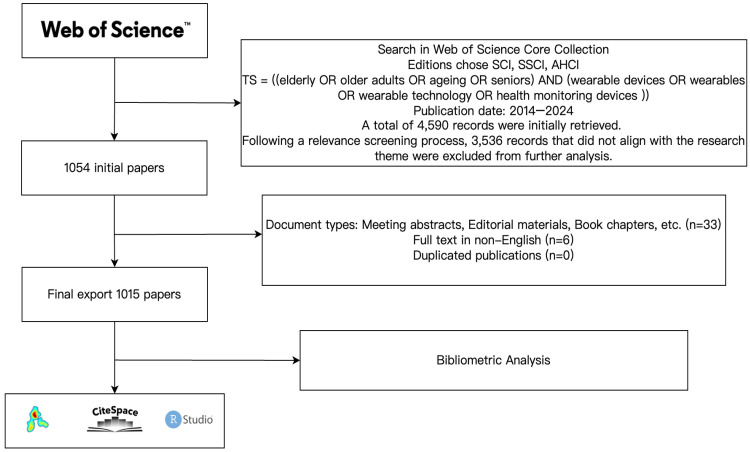
Flowchart of the literature selection.

**Figure 2 healthcare-13-02066-f002:**
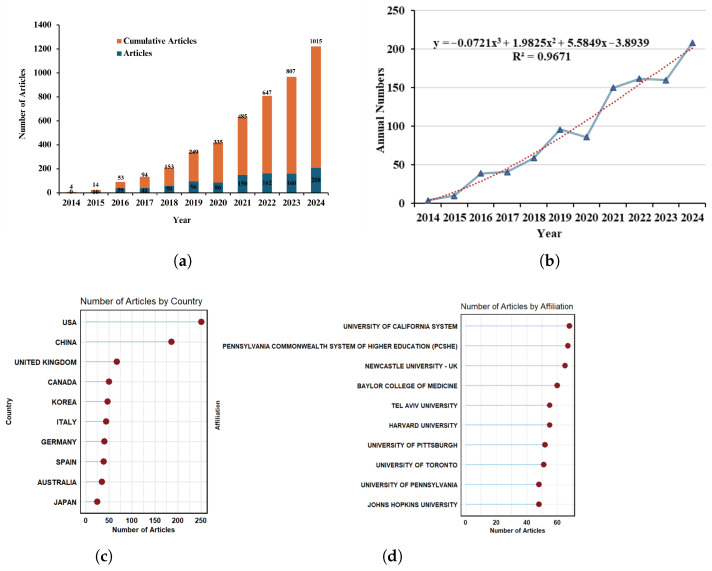
(**a**) Annual and cumulative number of publications from 2014 to 2024 and about the elderly (2014–2024). (**b**) Publication Trend. (**c**) Top 10 contributing countries in terms of publication output. (**d**) Top 10 contributing institutions in terms of publication output.

**Figure 3 healthcare-13-02066-f003:**
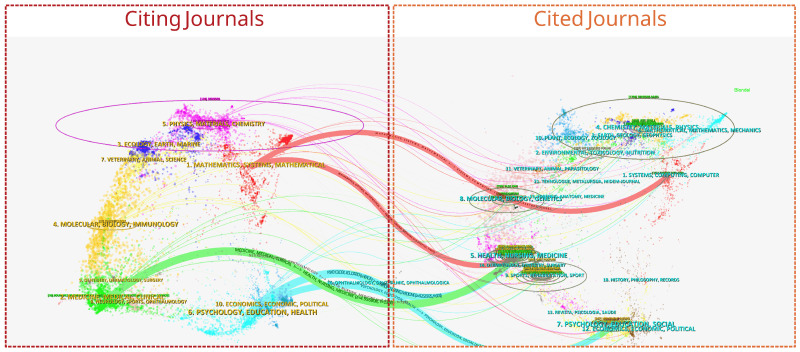
Double map overlay of journals.

**Figure 4 healthcare-13-02066-f004:**
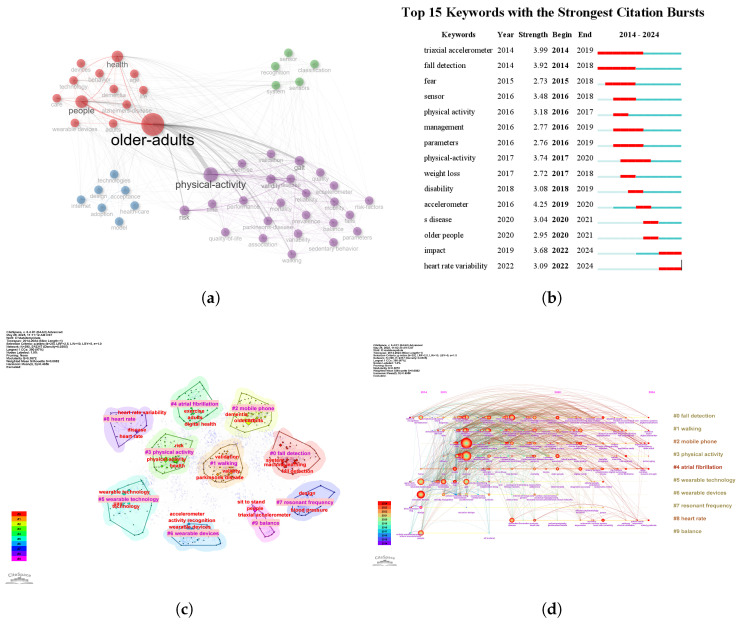
(**a**) Co-occurrence network of keywords. (**b**) Top 15 keywords with the strongest citation bursts (2014–2024). (**c**) Clustered map of keywords. (**d**) Timeline view of keyword clusters.

**Figure 5 healthcare-13-02066-f005:**
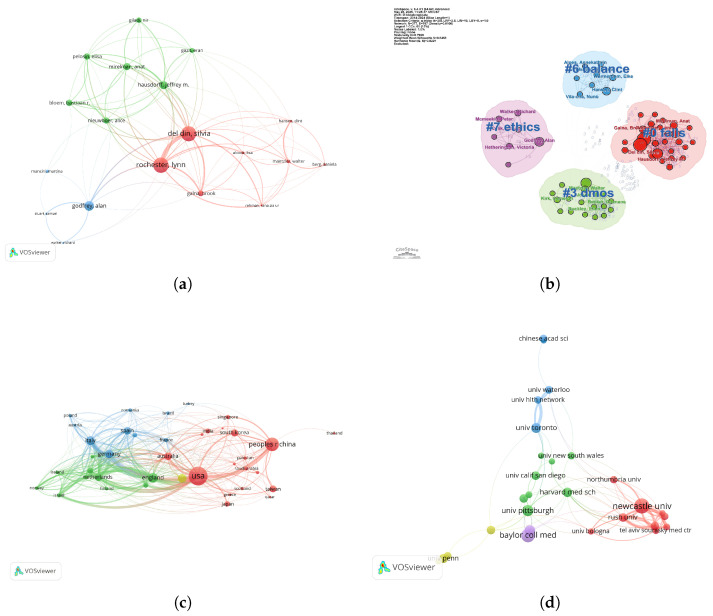
(**a**) Author collaboration network. (**b**) Author collaboration network clustering map. (**c**) Country-level collaboration network. (**d**) Institutional collaboration network.

**Figure 6 healthcare-13-02066-f006:**
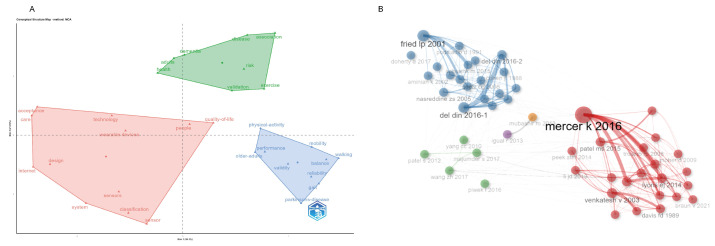
(**A**) Multiple Correspondence Analysis. (**B**) Co-citation network of references.

**Figure 7 healthcare-13-02066-f007:**
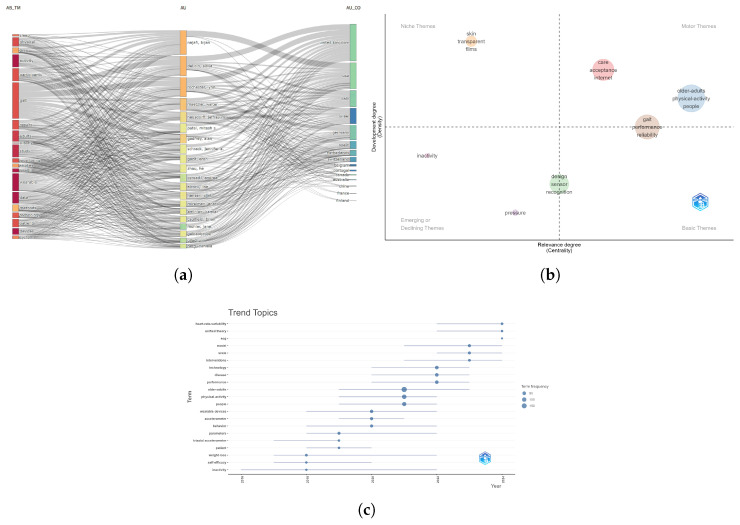
(**a**) Three-field plot connecting author keywords, authors, and countries. (**b**) Thematic map. (**c**) Trend topics map.

**Table 1 healthcare-13-02066-t001:** Top 10 contributing authors in the field of wearable devices for the elderly.

No.	Name	Affiliation	No. of Articles	Citations	H-Index
1	Bijan Najafi	University of California Los Angeles	29	738	49
2	Lynn Rochester	Newcastle University	22	1100	71
3	Silvia Del Din	Newcastle University	22	1080	35
4	Alan Godfrey	Northumbria University	13	639	34
5	Walter Maetzler	Schleswig Holstein University Hospital	12	199	61
6	Jeffrey Hausdorff	Rush University	10	504	110
7	Mitesh Patel	University of Pennsylvania	9	468	43
8	Jennifer Schrack	Johns Hopkins University	9	296	40
9	He Zhou	Shenzhen Dengding Biopharm Co Ltd	9	173	8
10	Anat Mirelman	Tel Aviv University	8	441	58

**Table 2 healthcare-13-02066-t002:** Most productive journals in the field of wearable devices for the elderly.

No.	Journals	Publisher	No. of Articles	Citations	5 Years Impact Factor	Impact Factor (2024)	Avg. Citations
1	SENSORS	MDPI	125	3236	3.7	3.5	25.888
2	JMIR MHEALTH AND UHEALTH	JMIR PUBLICATIONS, INC	37	2246	6.1	6.2	60.7027
3	JOURNAL OF MEDICAL INTERNET RESEARCH	JMIR PUBLICATIONS, INC	21	427	6.9	6.2	20.3333
4	GERONTOLOGY	KARGER	19	417	4	3	21.9474
5	IEEE ACCESS	IEEE-INST ELECTRICAL ELECTRONICS ENGINEERS INC	19	426	3.9	3.6	22.4211
6	IEEE SENSORS JOURNAL	IEEE-INST ELECTRICAL ELECTRONICS ENGINEERS INC	16	677	4.7	4.5	42.3125
7	ELECTRONICS	MDPI	15	245	2.6	2.6	16.3333
8	APPLIED SCIENCES-BASEL	MDPI	14	77	2.7	2.5	5.5
9	DIGITAL HEALTH	SAGE PUBLICATIONS LTD	12	99	3.7	3.3	8.25
10	HEALTHCARE	MDPI	12	133	2.8	2.7	11.0833

## Data Availability

No new data were created or analysed in this study. Data sharing is not applicable to this article.

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
