# Peer review of "Wearable Devices & Elderly: A Bibliometric Analysis of 2014–2024"

_healthcare, 2025, doi:10.3390/healthcare13162066_

Round 1

Reviewer 1 Report

Comments and Suggestions for Authors

The manuscript focuses on the field of ‘elderly population and wearable devices’ from 2014 to 2024, using bibliometric methods to sort out development trends, research hotspots, and global cooperation networks. This topic is in line with the needs of aging. The data is sourced from the Web of Science Core Collection, which is standardized and reliable. Tools such as CiteSpace are used to comprehensively present dimensions such as publication trends, keyword evolution, and international cooperation, revealing the transformation of research from ‘fall detection’ to ‘heart rate variability’ and ‘AI integration’. However, there are still some issues that require the authors to further improve or clarify. The specific comments are as follows:

Major Comments:

  1. The manuscript uses conventional bibliometric tools such as CiteSpace, VOSviewer, and R-Bibliometrix for analysis. These tools mainly focus on structural indicators of literature such as citation relationships, keyword co-occurrence, etc.,and do not introduce in-depth analysis of the literature content. So, please combine text mining, topic modeling, and other methods to conduct in-depth analysis of the literature content, identify potential topics and trends in the research, rather than relying solely on existing keywords or citation data.
  2. The core of the manuscript should focus on the research characteristics of ‘wearable devices dedicated to the elderly population’. However, a large portion of the Introduction such as content about AR/VR devices and smart headphones in ‘entertainment’ and ‘work scenarios’ deviates from the elderly population, excessively expanding to the generalized applications of wearable devices, which is inconsistent with the core topic. This deviation weakens the pertinence of the analysis and is also inconsistent with the research goal of ‘filling the gap in reviews in the elderly field’. So,the authors must focus on the core research object, delete redundant content unrelated to ‘wearable devices for the elderly population’, and strengthen the pertinence of the analysis.
  3. The current Discussion section only describes phenomena such as the migration of research topics and regional differences, but does not explain the specific guiding significance for clinical practice, industry, or policies in combination with elderly health needs, industrial bottlenecks, etc. Therefore,the authors could add new explanations in the Discussion to link the measurement results with practical issues, clarify the practical value of each trend, and avoid conclusions staying at the data level.

Minor Comments:

  1. Although the bibliometric method is adopted, the paperhas little discussion on the limitations of this method. It is recommended to add a critical discussion on the limitations of the bibliometric method and combine qualitative analysis to make up for the limitations of this method, especially the evaluation of literature quality.
  2. There may be biases in keyword selection. Section 2.1 of the manuscript shows that keyword selection depends on preset standards, which may ignore some emerging fields or related sub-topics. If possible, pleaseadopt more flexible keyword extraction methods to enhance the comprehensiveness and depth of the literature, or give a more comprehensive explanation of the rationality.
  3. On page 4, line 158, the cumulative number of publications by 2024 should be 1015, not 1051. The authorsshould check and revise it.
  4. On page 7, line 221, the attention years for the keyword ‘sensor’ are marked as ‘2015-2018’, which is inconsistent with ‘2016-2018’ shown in Figure 4(b). It needs to be verified and revised to ensure the accuracy of the content.

In conclusion, the reviewer hopes that the above comments will help the authors revise the manuscript and improve its value and clarity to readers.

Author Response

Dear Reviewer 1.

We would like to express our sincere appreciation for your careful reading of our manuscript and your valuable, constructive feedback. Your thoughtful comments have been instrumental in helping us improve the clarity, rigour, and overall quality of the paper.

Below, we present our point-by-point responses to each of your suggestions. To support your review, we have provided a tracked version, in which all revisions are highlighted in yellow.

Thank you again for the insightful suggestions.

Best regards,

Haojun Zhi, Mariia Zolotova

Comments and Response

Comment 1: The manuscript focuses on the field of ‘elderly population and wearable devices’ from 2014 to 2024, using bibliometric methods to sort out development trends, research hotspots, and global cooperation networks. This topic is in line with the needs of aging. The data is sourced from the Web of Science Core Collection, which is standardized and reliable. Tools such as CiteSpace are used to comprehensively present dimensions such as publication trends, keyword evolution, and international cooperation, revealing the transformation of research from ‘fall detection’ to ‘heart rate variability’ and ‘AI integration’.

However, there are still some issues that require the authors to further improve or clarify. The specific comments are as follows (Major Comments):

The manuscript uses conventional bibliometric tools such as CiteSpace, VOSviewer, and R-Bibliometrix for analysis. These tools mainly focus on structural indicators of literature such as citation relationships, keyword co-occurrence, etc.,and do not introduce in-depth analysis of the literature content. So, please combine text mining, topic modeling, and other methods to conduct in-depth analysis of the literature content, identify potential topics and trends in the research, rather than relying solely on existing keywords or citation data.

Response: We are grateful for your valuable suggestion, which has greatly contributed to improving the depth of our analysis. In response to this comment, we have incorporated a more detailed examination of the literature content in the following sections:

  1. In-depth analysis of top-cited publications – We have provided a qualitative and thematic examination of the Top 10 most cited publications in the field of wearable devices for the elderly (Table 3), offering insights into their research focus and contributions to the field. This discussion is presented on page 13, lines 415–445.
  2. Discussion of research themes and trends – In the Discussion section, we have expanded our analysis to identify and elaborate on key thematic directions, reflecting both established research hotspots and emerging topics. This discussion can be found on page 17, lines 594–667.
  3. Acknowledgement of methodological limitations – In the Limitations section, we explicitly recognise that our current approach did not employ advanced text-mining or topic-modelling techniques such as Latent Dirichlet Allocation (LDA). We also note that your recommendation offers valuable guidance for future research, which may integrate these methods to uncover latent thematic structures. This acknowledgement is presented on page 21, lines 793–799.

We believe these revisions address the concern by enhancing the interpretative depth of the bibliometric findings and by situating them within a broader thematic and methodological context, while also outlining a clear pathway for methodological enrichment in future studies.

Comment 2: The core of the manuscript should focus on the research characteristics of ‘wearable devices dedicated to the elderly population’. However, a large portion of the Introduction such as content about AR/VR devices and smart headphones in ‘entertainment’ and ‘work scenarios’ deviates from the elderly population, excessively expanding to the generalized applications of wearable devices, which is inconsistent with the core topic. This deviation weakens the pertinence of the analysis and is also inconsistent with the research goal of ‘filling the gap in reviews in the elderly field’. So,the authors must focus on the core research object, delete redundant content unrelated to ‘wearable devices for the elderly population’, and strengthen the pertinence of the analysis.

Response: We appreciate this constructive observation. In response, we have carefully revised the Introduction to ensure that it remains focused on the core research object, namely wearable devices dedicated to the elderly population. Specifically:

  1. All content unrelated to elderly-focused wearable devices, including sections on AR/VR devices and smart headphones in ‘entertainment’ and ‘work scenarios’, has been deleted.
  2. The revised Introduction now provides a more concise and targeted description of the research background.
  3. These changes can be found on page 2, lines 29–92.

We believe these revisions strengthen the pertinence and thematic coherence of the manuscript, aligning it more closely with the stated research objectives.

Comment 3: The current Discussion section only describes phenomena such as the migration of research topics and regional differences, but does not explain the specific guiding significance for clinical practice, industry, or policies in combination with elderly health needs, industrial bottlenecks, etc. Therefore,the authors could add new explanations in the Discussion to link the measurement results with practical issues, clarify the practical value of each trend, and avoid conclusions staying at the data level.

Response: We fully agree with the reviewer’s suggestion regarding the need to strengthen the practical relevance of the Discussion. In response, we have substantially revised this section to explicitly link the bibliometric findings with practical issues in elderly health management, industrial development, and policy formulation. Specifically:

  1. We have added explanations of how each identified research trend (e.g., the shift from fall detection to heart rate variability and AI integration) can inform clinical practice, such as improving personalising rehabilitation plans for older adults.
  2. We have discussed the industrial implications, including how technological innovations may address current bottlenecks in usability, interoperability, and device adoption among elderly users.
  3. These additions can be found on page 17, lines 594–667.

We believe these revisions enhance the practical significance of the manuscript by ensuring that the bibliometric insights are directly connected to real-world applications, thereby avoiding conclusions that remain solely at the data description level.

Comment 4: (Minor Comments)

Although the bibliometric method is adopted, the paper has little discussion on the limitations of this method. It is recommended to add a critical discussion on the limitations of the bibliometric method and combine qualitative analysis to make up for the limitations of this method, especially the evaluation of literature quality.

Response: We appreciate the reviewer’s valuable suggestion. In response, we have expanded the Limitations section to provide a more critical reflection on the methodological constraints of bibliometric analysis. Specifically, we have:

  1. Discussed the inherent limitations of relying solely on bibliometric indicators, including potential biases arising from database selection, citation-based metrics, and the exclusion of non-indexed literature.
  2. Suggested combining bibliometric analysis with qualitative methods in future research to address these shortcomings.

These additions can be found on page 21, lines 775–832. We believe these revisions address the reviewer’s concern and provide a more balanced and critical methodological discussion.

Comment 5: There may be biases in keyword selection. Section 2.1 of the manuscript shows that keyword selection depends on preset standards, which may ignore some emerging fields or related sub-topics. If possible, please adopt more flexible keyword extraction methods to enhance the comprehensiveness and depth of the literature, or give a more comprehensive explanation of the rationality.

Response: We thank the reviewer for this insightful observation. In Section 2.1 (page 3, lines 118–126), we have provided a clearer explanation of the keyword extraction process, outlining the rationale for using preset standards to balance recall and precision. Furthermore, as suggested, we have critically discussed the potential limitations of this approach in the Limitations section (page 21, lines 785–792), noting that fixed keyword strategies may overlook emerging topics or niche sub-fields. We have also emphasised that future studies could adopt more flexible or automated keyword extraction techniques—such as natural language processing-based approaches—to improve comprehensiveness and minimise selection bias (page 22, line 833-837).

Comment 6: On page 4, line 158, the cumulative number of publications by 2024 should be 1015, not 1051. The authors should check and revise it.

Response: We thank the reviewer for pointing out this numerical inaccuracy. The cumulative number of publications by 2024 has been corrected from 1051 to 1015 in the revised manuscript (page 6, line 220).

Comment 7: On page 7, line 221, the attention years for the keyword ‘sensor’ are marked as ‘2015-2018’, which is inconsistent with ‘2016-2018’ shown in Figure 4(b). It needs to be verified and revised to ensure the accuracy of the content.

Response: We appreciate the reviewer’s careful observation. The attention years for the keyword ‘sensor’ have been corrected from 2015–2018 to 2016–2018 to ensure consistency with Figure 4(b) (page 9, line 295 in the revised manuscript).

Comment 8: In conclusion, the reviewer hopes that the above comments will help the authors revise the manuscript and improve its value and clarity to readers.

Response: We sincerely thank the reviewer for the constructive and insightful comments. We have carefully addressed each point raised and revised the manuscript accordingly to enhance its academic value, clarity, and relevance to the readership. We believe that the changes made have substantially improved the quality of the work, and we appreciate the reviewer’s valuable guidance throughout this process.

Reviewer 2 Report

Comments and Suggestions for Authors

This is an interesting topic. However, a few concerns, especially methodological ones, should be addressed carefully.

  1. Line 116: According to many studies focusing on these bibliographic databases, Scopus is also a widely used database. Disclose this point to readers and justify your choice.
  2. Line 116: According to "The data source of this study is Web of Science Core Collection? Not enough" published in the established journal Scientometrics, the sub-datasets and coverage years of your used WoSCC MUST be disclosed for the sake of reproducibility. Disclose this point to readers with explanations.
  3. Line 117: The search field(s) should be given.
  4. Line 139: Pay attention to the problem of ambiguity in names.
  5. Line 140-141: Your data here doesn't support "increasing global cooperative nature".
  6. Line 154-156: Some studies published in Scientometrics also documented the expansion of Web of Science. This point should be added to partly explain the growth of publications.
  7. Table 1: The latest affiliation of these authors should be given.
  8. The main publishing outlets (journals) should be given.
  9. A limitation section should be added. For example, the regional bias of Web of Science is documented in "Regional disparities in Web of Science and Scopus journal coverage".
  10. It will also be helpful to add a funding analysis section. Besides, the limitations of funding information in Web of Science, as documented by many database experts, should be noted.

Author Response

Dear Reviewer 2.

We would like to express our sincere appreciation for your careful reading of our manuscript and your valuable, constructive feedback. Your thoughtful comments have been instrumental in helping us improve the clarity, rigour, and overall quality of the paper.

Below, we present our point-by-point responses to each of your suggestions. To support your review, we have provided a tracked version, in which all revisions are highlighted in yellow.

Thank you again for the insightful suggestions.

Best regards,

Haojun Zhi, Mariia Zolotova

Comments and Response

Comment 1: This is an interesting topic. However, a few concerns, especially methodological ones, should be addressed carefully.

Line 116: According to many studies focusing on these bibliographic databases, Scopus is also a widely used database. Disclose this point to readers and justify your choice.

Response: We thank the reviewer for this important suggestion. In Section 2.1 (page 3, lines 112–115), we have added an explicit explanation for not including the Scopus database, outlining the rationale for focusing exclusively on the Web of Science Core Collection. Furthermore, we have discussed this database choice in the Limitations section (page 21, lines 775–784).

Comment 2: Line 116: According to "The data source of this study is Web of Science Core Collection? Not enough" published in the established journal Scientometrics, the sub-datasets and coverage years of your used WoSCC MUST be disclosed for the sake of reproducibility. Disclose this point to readers with explanations.

Response: We appreciate the reviewer’s observation. The search fields have been described in detail in Section 2.1 (page 3, lines 99–157) and are visually presented in Figure 1 (page 5) for clarity.

Comment 3: Line 117: The search field(s) should be given.

Response: We appreciate the reviewer’s observation. The search fields have been described in detail in Section 2.1 (page 3, lines 99–157) and are visually presented in Figure 1 (page 5) for clarity.

Comment 4: Line 139: Pay attention to the problem of ambiguity in names.

Response: We thank the reviewer for highlighting this point. We have revised the relevant description to address potential ambiguities in names. The updated text can be found on page 5, lines 199–202.

Comment 5: Line 140-141: Your data here doesn't support "increasing global cooperative

Response: We appreciate the reviewer’s observation. In light of this comment, we have removed the statement from the manuscript, as this conclusion would require further detailed calculations for adequate support. The original remark was intended as a general summary of the overall dataset, but as it does not directly affect the aims or findings of the study, we have deleted it to ensure accuracy and avoid overgeneralisation.

Comment 6: Line 154-156: Some studies published in Scientometrics also documented the expansion of Web of Science. This point should be added to partly explain the growth of publications.

Response: Thank you for this valuable suggestion. We have incorporated this point into the revised manuscript to provide a more comprehensive explanation for the observed growth in publications. Specifically, we have added a note referring to studies published in Scientometrics that documented the expansion of the Web of Science database as a contributing factor. This addition can be found on page 6, lines 214–219 of the revised manuscript.

Comment 7: Table 1: The latest affiliation of these authors should be given. The main publishing outlets (journals) should be given.

Response: We appreciate the reviewer’s constructive feedback. Following the suggestion, we re-analysed the dataset using VOSviewer to ensure the accuracy and completeness of the author information. In the revised manuscript, we have updated Table 1 to include the most recent affiliations of the top authors, and we have added a detailed description of these affiliations (page 10, lines 333–343).

Additionally, we have created a new Table 2 titled “Most productive journals in the field of wearable devices for the elderly”, which presents the main publishing outlets, accompanied by a descriptive analysis (page 11, lines 349–362).

Comment 8: A limitation section should be added. For example, the regional bias of Web of Science is documented in "Regional disparities in Web of Science and Scopus journal coverage".

Response: We thank the reviewer for this valuable suggestion. In the revised manuscript, we have added a comprehensive Limitations section (page 21, lines 774–836), which critically discusses methodological constraints, including the potential regional bias of the Web of Science database, as documented in the referenced literature. This addition also addresses other limitations, such as database selection, keyword extraction, and the exclusion of certain advanced analytical techniques, thereby providing a more balanced interpretation of the study’s findings.

Comment 9: It will also be helpful to add a funding analysis section. Besides, the limitations of funding information in Web of Science, as documented by many database experts, should be noted.

Response: We appreciate the reviewer’s suggestion. In the revised manuscript, we have addressed this point in the Limitations section (page 22, lines 806–816). While we acknowledge the potential value of funding analysis, we have also discussed the incomplete and inconsistent coverage of funding information in the Web of Science Core Collection, as documented by database experts. To avoid introducing bias or misleading interpretations, we did not conduct a funding analysis in this study. However, we have noted it as a promising direction for future research when more comprehensive funding data becomes available.

Reviewer 3 Report

Comments and Suggestions for Authors

The manuscript provides a bibliometric analysis of research into wearable devices for the elderly conducted over the past decade. It highlights notable trends, leading contributors and thematic shifts. However, substantial improvement is required in several areas to meet academic and practical standards:

  1. There is no clear rationale for selecting the 2014–2024 period. The authors should explain why they chose this particular decade. Was it due to technological advances, the availability of data, or a surge in research activity? Furthermore, situating this timeframe within the broader historical evolution of wearable health devices for the elderly would strengthen the manuscript.
  2. Although the current search strategy uses relevant terms relating to both the elderly and wearable devices, this approach may be too narrow, risking the omission of important developments within the field. To ensure a comprehensive dataset, I recommend using a more dynamic, data-driven method to construct the search query. For the purposes of this bibliometric analysis, all keywords from the initial set of retrieved articles should be extracted and analysed in detail, for example by grouping them by frequency and identifying thematic clusters. The results of this analysis would then be used to refine and expand the search query, enabling a subsequent round of literature retrieval to capture additional relevant terminology, device categories, functionalities and application contexts that were not included initially.
  3. While the paper discusses research themes and trends, it lacks a systematic classification of wearable devices used by the elderly, particularly with regard to their technical and functional characteristics. Significant depth would be added by including a taxonomy of device types (e.g. sensors, smartwatches, fall detectors and biosignal monitors) alongside a discussion of clinical validation, certification and accuracy in the reviewed studies. This is particularly important given that device reliability, usability and regulatory approval are central to their adoption in elderly care.
  4. Table 2 lists highly cited publications, but it also includes review articles that are not directly related to device applications for the elderly. This may distort the findings. The table should distinguish between general, review and elderly-focused research. Alternatively, separate analyses should be provided and the limitations of the current approach should be acknowledged in the discussion.
  5. Concerns have been raised about the robustness of the conclusions regarding author and country dominance, which are based solely on data from a single database (Web of Science). This may introduce geographical and disciplinary bias. It is therefore recommended that findings are cross-validated with those from other major scholarly databases (e.g. Scopus and PubMed) and that the potential effects of database selection are discussed. Furthermore, the analysis does not include information on self-citations and cross-citations among leading authors, nor does it examine the temporal distribution of publications by prolific contributors. Including such details would provide a more nuanced view of scholarly influence and research dynamics.

Figures and Visualizations

Figures 2, 4, 5 and 7 are currently difficult to interpret. They contain too much information, often combining multiple visualisations into a single, crowded image, which reduces clarity. Each visualisation should be presented separately with appropriate labelling and resolution to enhance readability. The Double Map Overlay (Figure 3) is an interesting addition, but it should be accompanied by clear explanations of its findings, pathways and practical implications. Authors must ensure that all figures are legible and meaningfully support the text.

Comments on the Quality of English Language

The manuscript is generally well written, adopting a clear academic tone and logical structure. However, some issues require attention.

  • There are occasional grammatical errors and awkward phrasing, particularly in longer sentences and technical passages.
  • Some sections, particularly the Methods and Results sections, contain overly dense sentences that would be clearer if split up.
  • Figure legends and table captions are sometimes insufficiently descriptive or lack context, which impairs reader comprehension.
  • Specific terms are used inconsistently (e.g. 'wearable devices' vs. 'wearables' vs. 'wearable technology').

Author Response

Dear Reviewer 3.

We would like to express our sincere appreciation for your careful reading of our manuscript and your valuable, constructive feedback. Your thoughtful comments have been instrumental in helping us improve the clarity, rigour, and overall quality of the paper.

Below, we present our point-by-point responses to each of your suggestions. To support your review, we have provided a tracked version, in which all revisions are highlighted in yellow.

Thank you again for the insightful suggestions.

Best regards,

Haojun Zhi, Mariia Zolotova

Comment 1: The manuscript provides a bibliometric analysis of research into wearable devices for the elderly conducted over the past decade. It highlights notable trends, leading contributors and thematic shifts. However, substantial improvement is required in several areas to meet academic and practical standards:

There is no clear rationale for selecting the 2014–2024 period. The authors should explain why they chose this particular decade. Was it due to technological advances, the availability of data, or a surge in research activity? Furthermore, situating this timeframe within the broader historical evolution of wearable health devices for the elderly would strengthen the manuscript.

Response: We thank the reviewer for this insightful comment. In the revised manuscript, we have provided a detailed explanation of the rationale for selecting the 2014–2024 period in Section 2.1 (page 3, lines 128–152). This decade was chosen because it represents a phase of significant technological advancement in wearable health devices—particularly in sensor integration, data processing, and artificial intelligence—alongside a marked increase in global research activity and societal attention to ageing populations.

Comment 2: Although the current search strategy uses relevant terms relating to both the elderly and wearable devices, this approach may be too narrow, risking the omission of important developments within the field. To ensure a comprehensive dataset, I recommend using a more dynamic, data-driven method to construct the search query. For the purposes of this bibliometric analysis, all keywords from the initial set of retrieved articles should be extracted and analysed in detail, for example by grouping them by frequency and identifying thematic clusters. The results of this analysis would then be used to refine and expand the search query, enabling a subsequent round of literature retrieval to capture additional relevant terminology, device categories, functionalities and application contexts that were not included initially.

Response: We appreciate the reviewer’s constructive suggestion. As this study just represents a first-round analysis of research trends, we adopted the search strategy described in Section 2.1 (page 3, lines 99–157) and discussed its rationale accordingly. We have also acknowledged in the Limitations section (page 21, lines 785–792 and 833–839) that the current approach may omit certain developments due to its predefined scope. As suggested, we have identified the reviewer’s recommended dynamic, data-driven keyword extraction and iterative refinement process as a valuable methodological enhancement for future research. This will enable the inclusion of emerging terminology, device categories, functionalities, and application contexts not captured in the initial search, thereby enhancing both the comprehensiveness and depth of subsequent bibliometric analyses.

Comment 3: While the paper discusses research themes and trends, it lacks a systematic classification of wearable devices used by the elderly, particularly with regard to their technical and functional characteristics. Significant depth would be added by including a taxonomy of device types (e.g. sensors, smartwatches, fall detectors and biosignal monitors) alongside a discussion of clinical validation, certification and accuracy in the reviewed studies. This is particularly important given that device reliability, usability and regulatory approval are central to their adoption in elderly care.

Response: We thank the reviewer for this insightful suggestion. In response, we have added a systematic classification of wearable devices for the elderly in the Discussion section (page 17, lines 594–667). This taxonomy now explicitly categorises devices into sensors, smartwatches and wristbands, fall detectors, and biosignal monitors, highlighting their respective technical and functional characteristics. Furthermore, we have incorporated a discussion of clinical validation, certification requirements, and accuracy considerations reported in the reviewed studies, with particular emphasis on their relevance to device reliability, usability, and regulatory approval. These additions aim to strengthen the manuscript’s practical relevance and provide a clearer link between bibliometric trends and the real-world adoption of wearable technologies in elderly care.

Comment 4: Table 2 lists highly cited publications, but it also includes review articles that are not directly related to device applications for the elderly. This may distort the findings. The table should distinguish between general, review and elderly-focused research. Alternatively, separate analyses should be provided and the limitations of the current approach should be acknowledged in the discussion.

Response: We appreciate the reviewer’s valuable comment. Following this suggestion, we have re-analysed the dataset using VOSviewer and conducted a detailed reading of the highly cited publications. As a result, we have revised the table (now Table 3) to include only those works directly relevant to wearable device applications for the elderly (page 12, lines 415–445). Additionally, we have acknowledged in the Limitations section (page 22, lines 817–826) the potential constraints of this selection method and its influence on the interpretation of findings.

Comment 5: Concerns have been raised about the robustness of the conclusions regarding author and country dominance, which are based solely on data from a single database (Web of Science). This may introduce geographical and disciplinary bias. It is therefore recommended that findings are cross-validated with those from other major scholarly databases (e.g. Scopus and PubMed) and that the potential effects of database selection are discussed.

Response: We appreciate the reviewer’s insightful observation. As noted in Section 2.1 (page 3, lines 108–115), this study employed the Web of Science Core Collection due to its rigorous indexing standards, authority, and standardisation, which align with the study’s objectives. We acknowledge that relying on a single database may introduce geographical and disciplinary bias, and this limitation has been explicitly discussed in the Limitations section (page 21, lines 775–784). In future research, we intend to cross-validate our findings with data retrieved from other major databases, such as Scopus and PubMed, to enhance the robustness and comprehensiveness of the results.

Comment 6: Furthermore, the analysis does not include information on self-citations and cross-citations among leading authors, nor does it examine the temporal distribution of publications by prolific contributors. Including such details would provide a more nuanced view of scholarly influence and research dynamics.

Response: We thank the reviewer for this valuable suggestion. We fully agree that analysing self-citations, cross-citations among leading authors, and the temporal publication patterns of prolific contributors could yield additional insights into scholarly influence and collaboration dynamics. However, this study represents the first stage of our research and was designed to focus on aggregated trends at the journal and author levels and thematic hotspots, without extending to citation network-level analyses. This important point has been acknowledged in the Limitations section (page 22, lines 833–839) as a promising direction for future work, where such analyses will be incorporated to provide a more comprehensive understanding of research dynamics in this field.

Comment 7: Figures and Visualizations

Figures 2, 4, 5 and 7 are currently difficult to interpret. They contain too much information, often combining multiple visualisations into a single, crowded image, which reduces clarity. Each visualisation should be presented separately with appropriate labelling and resolution to enhance readability.

Response: We thank the reviewer for this valuable feedback. In the revised manuscript, Figures 2, 4, 5 and 7 have been re-uploaded and re-arranged to improve clarity and readability. Each visualisation is now presented separately with enhanced resolution and clear labelling, following the MDPI formatting guidelines. The updated figures are shown on pages 6, 8, 10, and 15, respectively.

Comment 8: The Double Map Overlay (Figure 3) is an interesting addition, but it should be accompanied by clear explanations of its findings, pathways and practical implications. Authors must ensure that all figures are legible and meaningfully support the text.

Response: We thank the reviewer for the constructive comment. In the revised manuscript, we have supplemented the description of the Double Map Overlay in detail (page 7, lines 241–277), providing a clearer explanation of its findings, citation pathways, and practical implications. This ensures that the figure meaningfully supports the text. Additionally, we have enhanced the resolution and labelling of Figure 3 to improve its legibility, in line with MDPI formatting requirements.

Comment 9: Comments on the Quality of English Language

The manuscript is generally well written, adopting a clear academic tone and logical structure. However, some issues require attention.

There are occasional grammatical errors and awkward phrasing, particularly in longer sentences and technical passages.

Some sections, particularly the Methods and Results sections, contain overly dense sentences that would be clearer if split up.

Figure legends and table captions are sometimes insufficiently descriptive or lack context, which impairs reader comprehension.

Specific terms are used inconsistently (e.g. 'wearable devices' vs. 'wearables' vs. 'wearable technology').

Response: We appreciate the reviewer’s constructive feedback on the language quality. We have carefully reviewed and revised the manuscript to address these issues. Specifically:

  • Corrected grammatical errors and improved awkward phrasing, especially in technical passages.
  • Split overly long sentences in the Methods and Results sections to enhance clarity and readability.
  • Expanded figure legends and table captions to provide sufficient context and aid comprehension.
  • Standardised terminology throughout the manuscript, consistently using “wearable devices” to maintain clarity and coherence.

These revisions improve both the logical flow and the overall readability of the manuscript.

Round 2

Reviewer 2 Report

Comments and Suggestions for Authors

This manuscript has been improved. However, some minor concerns, especially methodological ones, should be addressed carefully.

  1. Line 99-101: References should be added, especially those published in the last 5 years focusing on Web of Science Core Collection in established outlets such as Scientometrics and Quantitative Science Studies.
  2. Line 102-103: Your description of Web of Science is outdated. Replace the references in the materials and methods sections with authoritative studies in established journals.
  3. Line 106: Replace reference 24 with an authoritative one focusing on the database.
  4. Line 124-125: It is good to disclose the sub-datasets used in your study. The coverage years of the used datasets should also be disclosed, as advocated by a study in Scientometrics in 2019. This point should be explained to readers for responsible research.
  5. For the methods section, authoritative studies or original studies should be added to replace some unsuitable ones.
  6. Line 146-147: Please also mention the publication dates problem in Web of Science as probed by related studies.
  7. Line 218: Reference 23 is not suitable; replace it with some latest studies in Scientometrics focusing on the expansion of Web of Science Core Collection.
  8. Table 2: The year version of the impact factor should be given.
  9. Line 808-810: Related references should be added.
  10. Regional bias and non-English publications bias in Web of Science Core Collection, as probed by some studies in Scientometrics and Learned Publishing, should be mentioned as limitations.

Author Response

Dear Reviewer 2.

We would like to express our sincere appreciation for your careful reading of our manuscript and your valuable, constructive feedback. Your thoughtful comments have been instrumental in helping us improve the clarity, rigour, and overall quality of the paper.

Below, we present our point-by-point responses to each of your suggestions. To support your review, we have provided a tracked version, in which all revisions are highlighted in yellow.

Thank you again for the insightful suggestions.

Best regards,

Haojun Zhi, Mariia Zolotova

Comment 1: This manuscript has been improved. However, some minor concerns, especially methodological ones, should be addressed carefully.

Line 99-101: References should be added, especially those published in the last 5 years focusing on Web of Science Core Collection in established outlets such as Scientometrics and Quantitative Science Studies.

Response: We appreciate the reviewer’s valuable suggestion. We have added recent and authoritative references to strengthen the methodological basis of the section. The revisions can be found on page 3, lines 97–99.

Comment 2: Line 102-103: Your description of Web of Science is outdated. Replace the references in the materials and methods sections with authoritative studies in established journals.

Response: Thank you for this constructive suggestion. We have updated the description of the Web of Science in Section 2.1 (page 3, lines 99–104) to ensure it reflects the most current information.

Comment 3: Replace the references in the materials and methods sections with authoritative studies in established journals.

Response: We sincerely thank the reviewer for highlighting this important point. Following your recommendation, we have replaced the less suitable references in the Materials and Methods section with authoritative studies published. These updated references now provide stronger methodological support and ensure that the section aligns with the best practices in the field. The update can be found in sections 2.1 and 2.2 of the revised manuscript.

Comment 4: Line 106: Replace reference 24 with an authoritative one focusing on the database.

Response: We appreciate this suggestion. In the revised manuscript, we have replaced the previous Reference 24 with an authoritative publication from Quantitative Science Studies. This ensures the citation is both current and relevant to the methodological context of our study. The update can be found in Section 2.1 (page 3, line 106) of the revised manuscript.

Comment 5: Line 124-125: It is good to disclose the sub-datasets used in your study. The coverage years of the used datasets should also be disclosed, as advocated by a study in Scientometrics in 2019. This point should be explained to readers for responsible research.

Response: We thank the reviewer for highlighting the importance of disclosing dataset coverage years. This information was already included in Line 124 (“The search was conducted on May 28, 2025, and encompassed all publications from 2014 to 2024”). To make it more explicit, we have now integrated the coverage years with the description of the sub-datasets used, as recommended. The update can be found in Section 2.1 (page 3, line 127-129) of the revised manuscript.

Comment 6: For the methods section, authoritative studies or original studies should be added to replace some unsuitable ones.

Response: Thank you for this valuable suggestion. In the revised manuscript, we have carefully reviewed the references cited in the Methods section and replaced several that were less relevant with more authoritative and original studies from established journals. These updated references better align with the methodological framework of our study and strengthen the academic rigour of this section. The changes can be found in Section 2.2 (page 4, lines 166–195) of the revised manuscript.

Comment 7: Line 146-147: Please also mention the publication dates problem in Web of Science as probed by related studies.

Response: We appreciate the reviewer’s observation. In the revised manuscript, we have added a discussion on the publication date discrepancies in the Web of Science Core Collection, as identified in related scientometric studies. This addition appears in Section 2.1 (page 4, lines 150–153) and highlights the potential impact of such discrepancies on bibliometric analyses, ensuring readers are aware of this methodological consideration.

Comment 8: Line 218: Reference 23 is not suitable; replace it with some latest studies in Scientometrics focusing on the expansion of Web of Science Core Collection.

Response: We thank the reviewer for this suggestion. The cited reference has been replaced. This update appears on page 6, line 224 of the revised manuscript.

Comment 9: Table 2: The year version of the impact factor should be given.

Response: We appreciate the reviewer’s suggestion. In the revised manuscript, we have added the 2024 Journal Impact Factor for each listed journal in Table 2 to ensure clarity and temporal accuracy. This update is reflected on page 11, Table 2, with a note specifying that the values correspond to the 2024 release.

Comment 10: Line 808-810: Related references should be added.

Response: We have incorporated relevant references from previous bibliometric and information science research to support the statement. These citations are now included on page 22, lines 826 of the revised manuscript.

Comment 11: Regional bias and non-English publications bias in Web of Science Core Collection, as probed by some studies in Scientometrics and Learned Publishing, should be mentioned as limitations.

Response: We thank the reviewer for this valuable comment. In the revised manuscript, we have added a discussion of both regional bias and non-English publication bias in the Web of Science Core Collection, referencing relevant studies from Scientometrics and Learned Publishing. This addition appears in the Limitation section (page 22, lines 841–851) to acknowledge how these biases may affect the comprehensiveness and representativeness of our findings.

Comment 12: The English could be improved to more clearly express the research.

Response: We sincerely appreciate the reviewer’s observation regarding the clarity of the English expression. Following this suggestion, we have carefully revised the manuscript to improve readability, coherence, and logical flow. We have carefully reviewed and revised the manuscript to address these issues. Specifically:

  • Replaced “office work” with “workplace applications” (Page 1, line 22).
  • Revised the final sentence of the first paragraph for greater clarity (Page 1, lines 25–28).
  • Improved sentence structure in subsequent sections (Page 1, lines 31–34; Page 2, lines 81–84).
  • Added transitional sentences in Section 3.4 (Page 9, line 287, lines 299–301), Section 3.5 (Page 11, lines 379–381), Section 3.6 (Page 12, lines 388-389, lines 393-396, lines 418–420), and Section 3.7 (Page 14, lines 475–478) to enhance logical connections.
  • Corrected grammatical errors and eliminated confusing phrasing throughout.
  • Standardised terminology, consistently using “wearable devices” instead of mixed terms (“wearables”, “wearable technology”), thereby enhancing clarity and coherence.

These revisions improve both the logical flow and the overall readability of the manuscript.

Reviewer 3 Report

Comments and Suggestions for Authors

Thank you for your detailed and thoughtful responses to my review. I appreciate the time and effort you have dedicated to addressing each of my comments and suggestions.

In my assessment, the revisions you have introduced have significantly improved the clarity, structure, and overall quality of your manuscript. The additional explanations, expanded discussions, refined figures, and enhanced methodological transparency meaningfully strengthen the work and increase its value to the research community.

Author Response

Comment: Thank you for your detailed and thoughtful responses to my review. I appreciate the time and effort you have dedicated to addressing each of my comments and suggestions.

In my assessment, the revisions you have introduced have significantly improved the clarity, structure, and overall quality of your manuscript. The additional explanations, expanded discussions, refined figures, and enhanced methodological transparency meaningfully strengthen the work and increase its value to the research community.

Response: We sincerely thank the reviewer for their encouraging feedback and recognition of our efforts. We are pleased to hear that the revisions have improved the manuscript’s clarity, structure, and overall quality. We greatly appreciate your constructive comments and suggestions, which have been instrumental in strengthening the methodological transparency and practical value of our work.